# Iron Bioaccessibility and Speciation in Microalgae Used as a Dog Nutrition Supplement

**DOI:** 10.3390/vetsci10020138

**Published:** 2023-02-10

**Authors:** Thomas Dalmonte, Carla Giuditta Vecchiato, Giacomo Biagi, Micaela Fabbri, Giulia Andreani, Gloria Isani

**Affiliations:** Department of Veterinary Medical Sciences, Alma Mater Studiorum—University of Bologna, Via Tolara di Sopra 50, 50055-Ozzano dell’Emilia, 40064 Bologna, Italy

**Keywords:** *Arthrospira platensis*, *Chlorella vulgaris*, *Haematococcus pluvialis*, *Phaedactylum tricornutum*, in vitro digestion, biochemical characterisation

## Abstract

**Simple Summary:**

In recent years, the microalga market has grown due to its interesting nutritional profile, characterised by high quality proteins, bioactive molecules, and essential trace elements. Therefore, it is of vital importance to study how different species of microalgae can be used in animal nutrition, assessing how they are digested and what percentage of valuable nutrients is really absorbed. Iron is an essential trace element and the availability of natural sources of this nutrient may be useful in dog nutrition. This study investigated the iron content in four microalgae (*Chlorella vulgaris*, *Arthrospira platensis*, *Haematoccocus pluvialis*, and *Phaeodactylum tricornutum*) with emphasis on the iron bioaccessibility in dogs. The present results show that, in *C. vulgaris*, a high percentage (30%) of iron was available to be absorbed by the canine gastrointestinal system, suggesting that this microalga could be used as valuable iron supplementation in dog nutrition.

**Abstract:**

*Chlorella vulgaris*, *Arthrospira platensis*, *Haematoccocus pluvialis*, and *Phaeodactylum tricornutum* are species of interest for commercial purposes due to their valuable nutritional profile. The aim of this study was to investigate the iron content in these four microalgae, with emphasis on their iron bioaccessibility assessed using an in vitro digestion system to simulate the process which takes place in the stomach and small intestine of dogs, followed by iron quantification using atomic absorption spectrometry. Furthermore, the extraction of soluble proteins was carried out and size exclusion chromatography was applied to investigate iron speciation. Significant differences (*p* < 0.004) in iron content were found between *C. vulgaris*, which had the highest (1347 ± 93 μg g^−1^), and *H. pluvialis*, which had the lowest (216 ± 59 μg g^−1^) iron content. *C. vulgaris*, *A. platensis*, and *H*. *pluvialis* showed an iron bioaccessibility of 30, 31, and 30%, respectively, while *P. tricornutum* showed the lowest bioaccessibility (11%). The four species analysed presented soluble iron mainly bound to proteins with high molecular mass ranging from >75 to 40 kDa. *C. vulgaris* showed the highest iron content associated with good bioaccessibility; therefore, it could be considered to be an interesting natural source of organic iron in dog nutrition.

## 1. Introduction

Approximately 30% of microalgae production worldwide is sold in the animal feed market [1]. Together, *Chlorella vulgaris* and *Arthrospira platensis* represent the most popular microalgae produced for commercial purposes [2]. They are known and appreciated for their nutritional profile, including highly digestible proteins with an elevated content of essential amino acids [3], dietary fibres, n3-PUFAs, macro (Ca, Mg, P) and trace (Fe, Zn, Cu, Se) elements, and other essential molecules, in particular vitamins D and E. Other interesting microalgae are *Haematococcus pluvialis* which, in the red phase, is considered the best natural source of astaxanthin [4] and the diatom *Phaeodactylum tricornutum* as a sustainable source of nutrients, especially eicosapentaenoic acid, fucoxanthin, and chrysolaminarin [5]. Exhaustive data regarding the complete biochemical and nutritional profile of these microalgae are scarce. Sandgruber et al. (2021) have recently reported the nutrient composition of 15 microalgae, including *A. platensis*, *C. vulgaris*, and *H. pluvialis*, concluding that, despite wide variations among their biochemical composition, they can be considered an interesting source of nutrients for human nutrition [6]. Actually, *A. platensis* is a cyanobacterium. However, it is usually included with the microalgae; therefore, the term microalgae will be used to include all four species studied.

To obtain beneficial effects, algal nutrients must be digested and absorbed by the intestinal epithelium (bioaccessibility). They finally enter the circulatory system, becoming accessible to the organism (bioavailability). Many studies have investigated the bioaccessibility of nutrients from microalgae, focusing mainly on phenolic compounds, chlorophylls, and carotenoids [7]. Far less research has dealt with the bioaccessibility of trace elements, especially iron. Iron performs essential functions in biological systems as a cofactor for many enzymes and is essential for the assembly of heme and ferrous sulfide FeS clusters. These enzymes are involved in energy generation, free radical detoxification, synthesis of prostaglandins, DNA, fatty acid synthesis, and signal transduction [8]. Contrasting results have been reported regarding iron bioaccessibility in microalgae. In vitro [9,10] and in vivo [11] studies have suggested that *A. platensis* and *C. vulgaris* could be considered interesting sources of bioaccessible iron. On the other hand, Muszynska et al. (2017) [12] investigated iron bioaccessibility from commercial supplements based on *C. vulgaris* used as a human dietary supplement and concluded that they are not a good source of this essential element. 

In vitro digestion models are widely used to study digestibility of individual ingredients as well as complete diets in both human and animal nutrition. Moreover, these in vitro digestion models have already been adopted to evaluate the digestibility of algal biomasses in both humans [13,14] and dogs [15], with the final purpose of investigating the effects which microalgae may exert on the intestinal microbiota in vitro.

To the best of the authors’ knowledge, studies investigating iron bioaccessibility in dog feed and supplements are lacking. The majority of commercial pet foods contain different sources of iron in order to help meet dietary requirements; meat meals, meat and bone meals, dicalcium phosphate, and ferrous sulfate are rich in iron while milk is not. Therefore, growing dogs are very susceptible to dietary iron deficiency, while inadequate dietary iron intake is rare in adult dogs that are fed balanced diets [16,17]. The main cause of iron deficiency in dogs is chronic blood loss [18]. However, iron deficiency (and severe anaemia) has occasionally been described in intestinal bowel disease in dogs, following chronic intestinal blood loss [19]. Highly available sources of iron could therefore be useful to supplement this trace metal in dogs.

The aim of this study was to determine the iron content of *A. platensis*, *C. vulgaris*, *H. pluvialis*, and *P. tricornutum* before and after incubation with artificial digestive juices under conditions which simulate the canine gastrointestinal tract to determine its bioavailability. Moreover, extracts of these four microalgae were obtained to investigate iron speciation.

## 2. Materials and Methods

### 2.1. In Vitro Digestion

Four commercial samples of microalgae, *Arthrospira platensis* (AP), *Chlorella vulgaris* (CV), *Haematococcus pluvialis* (HP), and *Phaeodactylum tricornutum* (PT) kindly offered by Micoperi Blue Growth (Ravenna, Italy) were submitted to a digestion in vitro to simulate the process that takes place in the stomach and small intestine of dogs as described by Biagi et al. (2016) [20]. Briefly, the samples were subjected to two phases of incubation, which occurred under controlled temperature (39 °C). The first one lasted two hours in the presence of pepsin, gastric lipase, and 0.075 N HCl followed by a second four-hour phase with phosphate-bicarbonate buffer pH 7.5, pancreatin, and bile salts. At the end, a supernatant representative of the digested bioaccessible fraction and a pellet representative of the undigested fraction were obtained. The pellets were dried at 65 °C until a constant dry weight was obtained. The supernatants were frozen and stored at −20 °C for further analyses. Microalgae samples were digested in triplicates. These supernatants and pellets were obtained from our previous study [15].

### 2.2. Iron Detection and Bioaccessibility

Algal biomasses, supernatants, and pellets obtained from in vitro digestion were analysed to determine iron content, using flame atomic absorption spectrometry (AAS) following the mineralization and analysis protocol reported by Isani et al. (2022a) [21]. The iron detection limit was 0.04 μg mL^−1^. The iron concentration is reported as μg g^−1^ dry weight (dw) or as μg mL^−1^ depending on the sample analysed. 

In this research, the term bioaccessibility is used to define the fraction of iron released from the algal biomass during the in vitro digestibility analysis performed by Delsante et al. (2022) [15]. To determine the bioaccessible iron fraction, the following equation was used:([Fe] supernatant/[Fe] algal biomass) × 100(1)

### 2.3. Extraction of Soluble Proteins

The extraction of soluble proteins was carried out starting from microalgal biomass. The alga biomass (100 mg) was pulverised and extracted following the protocol reported by Isani et al. (2022a) [21]. Iron concentration was measured as reported in Section 2.2. Extractions were performed in duplicate. 

### 2.4. Iron Speciation after Size Exclusion Chromatography (SEC)

A volume of 0.8 mL of supernatant obtained from each algal extract was applied to a Sephadex G-75 chromatography column (0.9 × 90 cm). A gel filtration marker kit (MWGF70-1KT, Sigma-Aldrich, St Louis, MO, USA) was used to calibrate the column. Fractions of 1.5 mL were collected and analysed for iron concentration as described in Section 2.2. In the supernatants and fractions obtained from SEC, total proteins, chlorophyll a, and phycocyanin were determined by measuring the absorbance at 280, 430, 662, and 620 nm, respectively (DeNovix DS-11 Series Spectrophotometer, Wilmington, DE, USA). In addition, in the samples from *A. platensis*, phycocyanin was detected at 620.

Furthermore, in the samples of *H. pluvialis*, astaxanthin was determined at 477 nm. 

### 2.5. Statistical Analysis

The central limit theorem was applied in order to choose the correct statistic test [22]. Thus, taking into consideration the number of samples for each microalga (*n* = 3), the Kruskal–Wallis one-way ANOVA with Dunn’s multiple comparisons post-hoc test was performed to determine significant differences of iron among the four microalgae, pellets, supernatants and bioaccessible fraction obtained by in vitro digestion, pellets, and supernatants obtained by soluble proteins extraction. The *p*-value obtained underwent to Bonferroni’s correction and differences between groups were considered significant for *p* < 0.05. 

Statistical analyses were performed using R 4.2.1 (R foundation for statistical computing; Vienna, Austria; https://www.R-project.org/, accessed on 1 October 2022). Data are reported as median, mean ± SD (standard deviation). All graphs were made using GraphPad Prism version 9.2 (GraphPad Software, San Diego, CA, USA).

## 3. Results and Discussion

### 3.1. Iron Content in Microalgae

The iron content in microalgae follows this decreasing order: *C. vulgaris* > *P. tricorutum > A. platensis > H. pluvialis*, and is reported in Figure 1. Significant differences were found between *C. vulgaris* and *H. pluvialis*. 

Of the four species examined, *C. vulgaris* had the highest iron content (1347 ± 93 μg g^−1^). The value was in the range of those reported by other authors. Muszynska et al. (2017) [12] analysed different commercial samples of *C. vulgaris* and found iron contents ranging from 880 to 2112 μg g^−1^ while Kejzar et al. (2021) [23] found a value of 1000 ± 520 μg g^−1^ in *Chlorella* spp. and Rzymski et al. (2019) [24] found values from 438 to 1661 μg g^−1^. On the other hand, Sandgruber et al. (2021) [6] reported an average iron content of 536 μg g^−1^ in two *C. vulgaris* samples while Maruyama et al. (1997) [25], Tokosoglu et al. (2003), [26] and Panahi et al. (2012) [27] reported iron contents equal to 2000, 2600, and 6800 μg g^−1^ in *C. vulgaris*, respectively. 

In the present study, *A. platensis* presented an iron content of 335 μg g^−1^, a value which was lower than those reported in the literature. Isani et al. (2022b) [28] reported that ten commercial samples of *A. platensis* had an iron content between 353 and 1459 μg g^−1^. Principe et al. (2020) [10] detected values between 63 and 1066 μg g^−1^ in samples of *A. platensis* from the Argentinian market. Kejzar et al. (2021) [23] reported iron values of 1360 ± 1330 μg g^−1^ and Rutar et al. (2022) [29] reported values from 370 to 3480 μg g^−1^. Finally, Sandgruber et al. (2021) [6] and Rzymski et al. (2019) [24] reported iron contents in *A. platensis* ranging from 11.6 to 835 μg g^−1^ and from 369 to 2287 μg g^−1^, respectively. 

Data regarding the iron content of the other two microalgae are limited. In the present study, *H. pluvialis* presented an iron content of 216 μg g^−1^, lower than the value of 972 μg g^−1^ reported by Sandgruber et al. (2021) [6] while *P. tricornutum* had a higher iron content of 956 ± 267 μg g^−1^. Although various studies [30,31] have investigated the response of this microalga grown in media with varying iron concentrations, to the authors’ knowledge, there are no studies reporting its iron content, likely because *P. tricornutum* cannot be marketed for human use. 

### 3.2. Iron Bioaccessibility 

The amount of iron released into the soluble fraction after gastrointestinal digestion is the bioaccessible fraction which has been calculated using Equation (1). Therefore, the bioaccessibility of iron in the four microalgae was determined using the iron content measured in the digested fractions obtained in an in vitro system simulating the process which takes place in the stomach and small intestine of dogs. 

The values obtained demonstrated a similar iron bioaccessibility of 31% for *A. platensis* and 30% for *C. vulgaris* and *H. pluvialis*, while *P. tricornutum* had the lowest bioaccessibility, namely 11% (Table 1). The interaction of different factors impacts iron bioaccessibility; these include the cellular organisation of the microalgae and the biochemical nature of iron ligands. In *A. platensis*, the high bioaccessibility associated with the highest digestibility could be due to the structure of Gram-negative bacterial cells which are surrounded by a wall of peptidoglycans, while the other eucaryotic algae have a cellulose cell wall which opposes more resistance to digestive enzymes. In particular, *H. pluvialis*, in the aplanospore form or the red phase, presents a rigid cell wall which can hinder the action of digestive enzymes, limiting access to the intracellular molecules [4,32]. Despite its low digestibility, *H. pluvialis* presented an iron bioaccessibility similar to *A. platensis* and *C. vulgaris*; however, the low iron content in the sample analysed (216 μg g^−1^ dw) resulted in the lowest iron bioaccessibility of the four microalgae (Table 1). On the contrary, despite its relatively high digestibility, this study disclosed that the diatom *P. tricornutum* had the lowest iron bioaccessibility of the four microalgae analysed. However, due to the high iron content in the algal sample (956 μg g^−1^ dw), one gram of this microalga contained 105 μg of bioaccessible iron. 

The minimum iron requirements of adult dogs according to the European Pet Food Industry federation (FEDIAF) (2021) [33] are 9 mg per 1000 kcal of metabolisable energy. Accordingly, considering that *C. vulgaris* provides 404 μg g^−1^ of bioaccessible iron, the daily iron requirement of an adult dog with a body weight of 10 kg (approximately 5.6 mg per day) could be covered by 14 g of *C. vulgaris*. 

Contrasting results have been reported in the literature regarding the bioaccessibility of iron from microalgae; this important issue is still a matter of debate. Puyfoulhoux et al. (2001) [9] examined the iron availability in *A. platensis* using an in vitro digestion/Caco-2 cell culture system, concluding that this microalga could represent an adequate source of iron. Principe et al. (2020) [10] reported an iron bioaccessibility ranging from 2.8 to 27% under oral conditions, 3 to 60% under gastric conditions, and 3 to 40% under intestinal conditions for *A. platensis* in different human digestion phases while Rutar et al. (2022) [29] reported that the iron bioavailability in *A. platensis* was low, due to the small amount of iron in the more bioavailable ferrous ion Fe^2+^. Uribe-Wandurraga et al. (2020) [34] reported an iron bioaccessibility from 40 to 50% in cookies enriched with 1.5 and 2% *C. vulgaris* and *A. platensis*, using an in vitro system which simulated the human digestive process. In 32 pregnant women (second and third trimester), the oral supplementation with *Chlorella pyrenoidosa* for 12–18 weeks decreased markers of anaemia as compared to the control group [11], suggesting that *C. pyrenoidosa* contained bioaccessible and bioavailable iron. On the contrary, Muszynska et al. (2017) [12] reported that the content of iron in commercial preparations of *C. vulgaris* was negligible after incubation with artificial digestive juices, indicating that the preparations examined were not a good source of this essential element.

### 3.3. Iron Speciation and Protein Fractioning Using Size Exclusion Chromatography (SEC)

Th various bioaccessibilities determined from the four microalgae analysed could also be related to the molecular characteristics of iron ligands. In biological systems, iron can bind to different ligands, either inorganic, such as ferrihydrite, or organic, such as iron-binding proteins. The speciation of the element, namely its distribution among these molecules, is an important issue when studying its bioaccessibility. Therefore, to shed more light on this important topic, the four microalgae were extracted to analyse the distribution of iron between the insoluble fraction (pellets), which, for the most part, contains inorganic iron, and the soluble fraction (supernatant), which contains iron bound to hydrophilic molecules, for the most part proteins, peptides, and amino acids. In all the samples analysed, the iron content was higher in the pellet than in the soluble fraction (Figure 2). The percentage of the iron content determined in *A. platensis* was in the range of data reported by Isani et al. (2022 b) [28] in the same species. The data in the present study provided additional evidence as to what had been hypothesised by Perfiliev et al. (2018) [35] that *A. platensis* accumulated iron, for the most part in an inorganic form such as ferrihydrite. An even higher iron content in pellets was found in *C. vulgaris*, *H. pluvialis*, and *P. tricornutum*, with percentages from 92 to 97%. Ferrihydrite could be responsible for the high iron percentages determined in the pellet fraction of these three microalgae.

Subsequently, to investigate the iron speciation in the soluble fractions, supernatants underwent SEC associated with sensitive metal detection in the fractions using atomic absorption spectroscopy (AAS), a hyphenated approach which is commonly used in metallomic studies [36] and had previously been applied to *A. platensis* [21,28]. Total proteins and iron measured in fractions after SEC are reported in Figure 3 and Figure 4. Phycocyanin in *A. platensis*; chlorophyll a in *A. platensis*, *C. vulgaris*, and *P. tricornutum*; and astaxanthin in *H. pluvialis* are reported in Figure 5, Figure 6 and Figure 7. 

In all the samples analysed, a major protein peak was eluted between fractions 11 and 15 (Figure 3). This peak contains proteins with a high molecular mass (HMM), ranging from >75 to 40 kDa. *A. platensis* showed a higher content of soluble proteins than the other microalgae in accordance with its high digestibility. A second peak of absorbance is present between fractions 29 and 34. Small peptides, amino acids and, finally, free ions (starting from fraction 32) elute in these chromatographic fractions. In *A. platensis*, it has been hypothesised that mycosporine-like amino acids (MAAs) may be present among these low molecular mass molecules [28]. The presence of a second peak is also detectable in *C. vulgaris*, *H. pluvialis*, and *P. tricornutum*, suggesting the presence of small peptides and free amino acids in these species too. Accordingly, Ba et al. (2016) [37] reported the presence of low molecular mass proteins (<10 kDa) in *H. pluvialis* in the vegetative phase after SEC protein separation. 

In all the samples analysed, the elution profile of iron after SEC presented a major peak between fractions 11 and 14 (Figure 4), overlapping the main peak of proteins at HMM (Figure 3), indicating that the most relevant metal burden is bound to proteins. *A. platensis* shows the highest amount of iron, followed by *C. vulgaris*, while *H. pluvialis* and *P. tricornutum* are characterised by a far lower iron concentration. Other minor peaks are present in fractions containing intermediate MM molecules from 40 to 20 kDa (fractions 15–18) and low MM molecules (fractions 28–32). In *A. platensis*, iron was also bound to ligands with an MM of 12–10 kDa (fractions 21–25).

The nature of these ligands is still an open question. In *A. platensis*, the elution profile of phycocyanin (Figure 5) overlaps both the total proteins and the iron peaks, with a maximum at fraction 12 (Figure 3 and Figure 4). Phycocianin is a multimeric blue phycobiliprotein containing α and β subunits. Phycocianin is involved in the photosynthetic machinery of *A. platensis*, functioning as a light-capturing antenna in photosystem II. In addition to this essential role, it has been reported by Bermejo et al. (2008) [38] that this protein was able to bind iron in vitro. The data in the present study add additional evidence regarding the role of phycocyanin as an iron binding protein in vivo as recently suggested by Isani et al. (2022a and b) [21,28]. Regarding *P. tricornutum*, Sutak et al. (2012) [39] reported the presence of ISIP1 (iron starvation-induced protein 1), suggesting a role of this protein in iron uptake. The ISIP1 protein has a molecular mass of 61 kDa, in the range of those eluted after SEC in fractions 10–15 in which a peak of proteins in the *P. tricornutum* supernatant and a peak of iron were found. Behnke et al. (2020) [40] confirmed that ISIP1 was an additional protein involved in the utilisation of diverse iron pools, thereby securing the success of *P. tricornutum* in iron-poor environments.

*A. platensis* showed the highest chlorophyll a content while *C. vulgaris* and *P. tricornutum* showed a similar but lower content. These three microalgae showed a unique peak between fractions 11 and 14. In *H. pluvialis*, no chlorophyll was detected. These data add additional proof that chlorophyll decreases when *H. pluvialis* starts to accumulate astaxanthin as reported by Fang et al. (2019) [41]. In *H. pluvialis*, astaxanthin shows a peak between fractions 11 and 14 (Figure 7). Despite its lipophylic nature, astaxanthin has been extracted, at least in part, in the soluble hydrophilic extract and, after SEC, appears in HMM chromatographic fractions, likely bound to proteins. 

## 4. Conclusions

To the best knowledge of the authors, this is the first study which investigated iron bioaccessibility from *A. platensis*, *C. vulgaris*, *P. tricornutum*, and *H. pluvialis* using a validated in vitro canine digestion simulation system. Of the four algal samples analysed, *C. vulgaris* contained the highest iron content associated with high bioaccessibility. Therefore, this microalga provided the highest amounts of iron available to be absorbed across the intestinal epithelium of dogs and could be considered as an interesting iron supplement in dog nutrition.

Taken together, the data in the present study suggested that the four species analysed presented soluble iron mainly bound to proteins with an HMM ranging from >75 kDa to 40 kDa, with emphasis on phycocyanin in *A. platensis*.

This study has provided a promising foundation for investigating the supplementation of microalgae in dog nutrition with the possibility of obtaining health benefits. Nevertheless, additional studies are needed to increase knowledge regarding the effects of microalgae supplementation in dogs and the pathways involved in iron speciation in microalgae.

## Figures and Tables

**Figure 1 vetsci-10-00138-f001:**
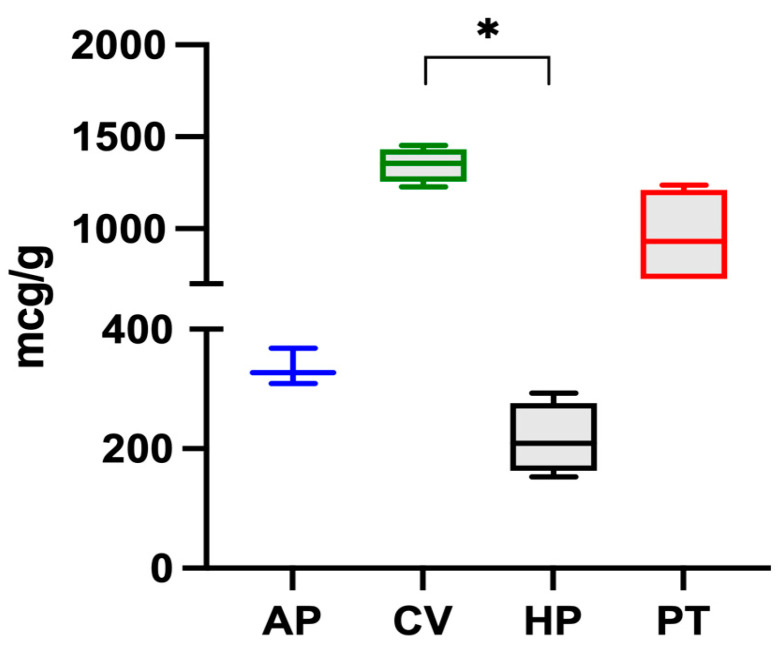
Iron content in the four microalgae (*A. platensis*, *C. vulgaris*, *H. pluvialis*, *P. tricornutum*) analysed. The data are expressed as micrograms per gram of dry algae biomass and are reported as medians and interquartile ranges. * Significantly different for the Dunn post-hoc test (*p* < 0.05).

**Figure 2 vetsci-10-00138-f002:**
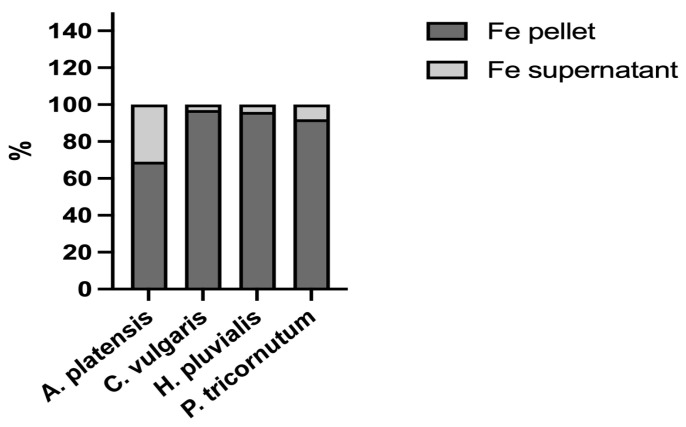
Iron distribution between the supernatants (soluble fractions) and pellets (insoluble fractions) obtained after the extraction of *A. platensis*, *C. vulgaris*, *H. pluvialis*, and *P. tricornutum*. The data are expressed as percentages.

**Figure 3 vetsci-10-00138-f003:**
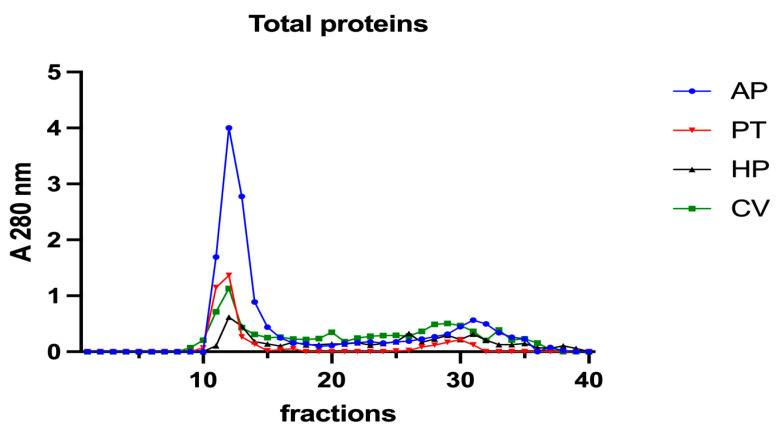
Chromatographic profiles of total proteins after SEC of extracts from algal samples. Proteins were detected at 280 nm. Each chromatographic profile is the mean of two chromatographies. AP = *A. platensis*; PT = *P. tricornutum*; HP = *H. pluvialis*; CV = *C. vulgaris*.

**Figure 4 vetsci-10-00138-f004:**
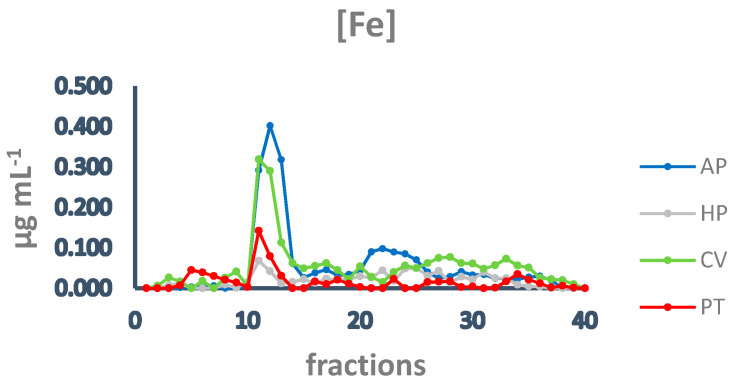
Chromatographic profiles of iron after SEC of extracts from algal samples. Iron concentration is expressed as µg mL^−1^. Each chromatographic profile is the mean of two chromatographies. AP = *A. platensis*; PT = *P. tricornutum*; HP = *H. pluvialis*; CV = *C. vulgaris*.

**Figure 5 vetsci-10-00138-f005:**
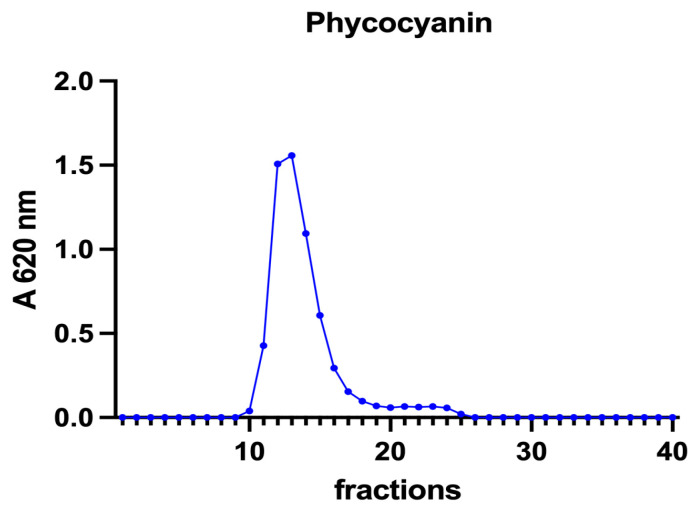
A chromatographic profile of phycocyanin after SEC of the *A. platensis* extract. Phycocyanin concentration is reported as mg mL^−1^. The chromatographic profile is the mean of two chromatographies.

**Figure 6 vetsci-10-00138-f006:**
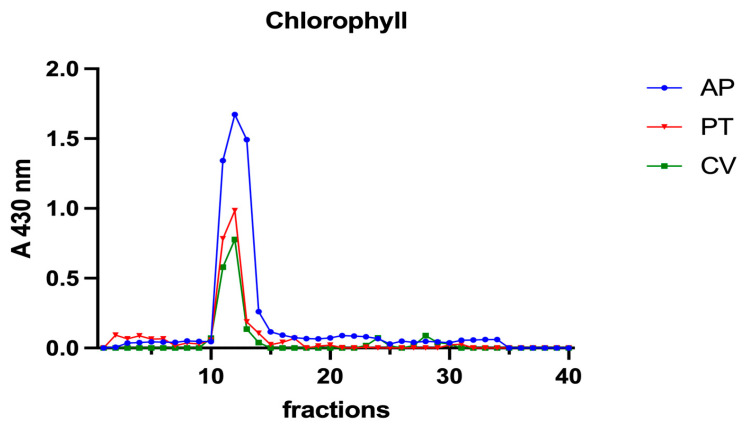
Chromatographic profiles of chlorophyll a after SEC of extracts from algal biomass. Chlorophyll a was detected at 430 nm. Each chromatographic profile is the mean of two chromatographies.

**Figure 7 vetsci-10-00138-f007:**
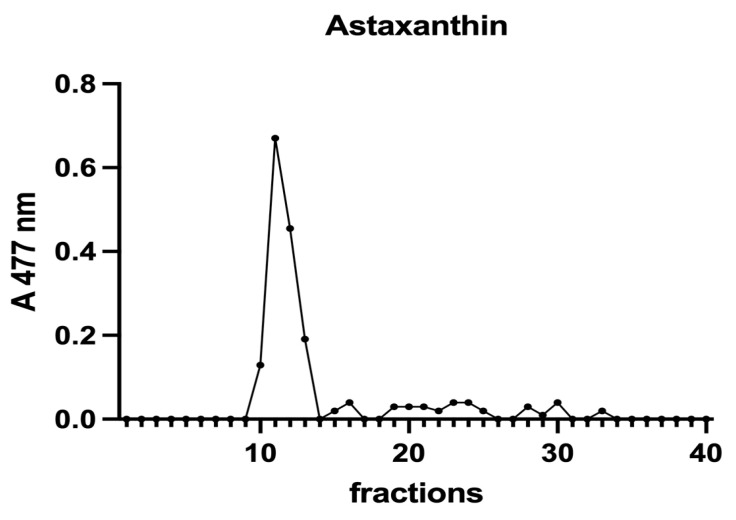
A chromatographic profile of astaxanthin after SEC of an *H. pluvialis* extract. Astaxanthin was detected at 477 nm. The chromatographic profile is the mean of two chromatographies.

**Table 1 vetsci-10-00138-t001:** The iron content in algal biomass, digestibility and bioaccessibility in four microalgae (*A. platensis*, *C. vulgaris*, *H. pluvialis*, *P. tricornutum*) undergoing an in vitro digestion process.

Algae	Digestibility ^§^ (%)	Iron DF * μg mL^−1^	Iron UF ° μg g^−1^ dw	Bioaccessible Fraction %	Bioaccessible Iron μg g^−1^ alga
*A. platensis*	86	1.30 ± 0.43	1955 ± 352 ^b^	31 ± 12	104 ± 9
*C. vulgaris*	55	5.04 ± 2.15 ^a^	1861 ± 267	30 ± 12	404 ± 28 ^c^
*H. pluvialis*	8	0.80 ± 0.15 ^a^	226 ± 49 ^b^	30 ± 9	64 ± 18 ^c^
*P. tricornutum*	68	1.31 ± 0.29	1690 ± 70	11 ± 0.6	105 ± 29

^§^ The data were obtained from Delsante et al. (2022) [15]. * DF: digested fractions (supernatants after in vitro digestion); ° UF: undigested fractions (pellets after in vitro digestion). In the same column, the same superscript letters indicate significant differences between the microalgae (*p* < 0.05). Data are expressed as mean ± standard deviation.

## Data Availability

Not applicable.

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
