# Peer review of "Iron Bioaccessibility and Speciation in Microalgae Used as a Dog Nutrition Supplement"

_vetsci, 2023, doi:10.3390/vetsci10020138_

Round 1

Reviewer 1 Report

The study is well designed, performed and reported. Iron deficiency is not common in dogs, but dietary iron supplementation is needed. Iron from micro algae might be a future natural source. Mineral absorption mechanisms is complicated in vivo and even more difficult to copy in vitro. You have shown that the iron content and iron bioavailibility is different among micro algae species which is important information for dietary inclusion level.    

Author Response

We want to thank the reviewer for the positive comment.

Reviewer 2 Report

The paper presents the properties of some microalgae with emphasis on iron content and the possibility to be used in dog nutrition. 

Please see reference style (L59 and the whole manuscript).

Introduction: I recommand discussing more the chemical composition of algae.

Methods: L74 I suggest to give more details: time, temperature, etc. 

Results and discussion:

Table 1: The standard deviation and statistics are missing. 

Figures: Please add statistic significances (letters) between samples. 

Overall, the study is very simple. Additional data could be included (e.g. FT-IR). 

Author Response

  • The paper presents the properties of some microalgae with emphasis on iron content and the possibility to be used in dog nutrition. Please see reference style (L59 and the whole manuscript).

The authors reformulated the reference style as requested in the whole manuscript.

  • Introduction: I recommand discussing more the chemical composition of algae.

Chemical and nutritional properties of three microalgae were implemented. To the knowledge of the authors. there are scarce information’s in bibliography regarding the biochemical composition of P. tricornutum.

  • Methods: L74 I suggest to give more details: time, temperature, etc. 

More details of the method published by Biagi et al. (2016) were added from L74 to L91

  • Results and discussion:

Table 1: The standard deviation and statistics are missing. 

Standard deviation and statistics outcomes were added in Table 1.

Figures: Please add statistic significances (letters) between samples. 

Statistics significances (letters) were added in Figure 1 and 2. Others Figures are chromatographic profiles; no statistics was used.

  • Overall, the study is very simple. Additional data could be included (e.g. FT-IR).

It is unclear to the authors what the reviewer means by FT-IR. Fourier-transform infrared spectroscopy? So, we could not meet the request.

Reviewer 3 Report

The manuscript ID: vetsci-2161942, entitled “Iron bioaccessibility and speciation in microalgae used as dog nutrition supplement“ is interesting, but I recommend to improve this rough draft to greater extent that somehow gain better citation game in later time after publication.

The Introduction section can be extended by including an adequately argumentation of the research goal.

Material section, especial Statistic analysis, need to be revised.

The equations from Lines 103 and 128 from Material section need to be

numbered and their numbers need to be specified in the manuscript.

The Statistical analysis section needs to be reformulated in order to be

clearer. See the example from Other remarks.

The results and the discussions need to be more detailed.

Please complete the Results and discussions section with more

information if the results obtained are in agreement/disagreement with

those from other recent studies.

Extensive English editing is required.

Other remarks:

Line 28-30: (P=0.004) need to be inserted immediately after “Significant differences“

Line 110:for 10 min at 38 kHz … instead of … for 10 minutes at 38 kHertz …

Line 131-132, 135-136: Please reformulate! For example: Kruskal–Wallis one-way ANOVA with Dunn’s multiple comparisons were performed for data with unequal variances. The normally distributed data were compared using a one-way ANOVA with Dunnett’s multiple comparison test.

Line 138-139: … using Rstudio and R 4.2.1 version run in console … Please reformulate!

Line 183, 193: Table 1 instead of table 1

Line 235: Figure 2 instead of figure 2

Line 246: Figures 3 and 4 instead of figures 3 and 4

Line 248: Figures 5, 6, and 7 instead of figures 5, 6, and 7

Line 257: Figure 3 instead of figure 3

Line 269: Figure 4 instead of figure 4

Line 281: Figure 5 instead of figure 5

Line 282: Figures 3 and 4 instead of figures 3 and 4

Line 300: Figure 7 instead of figure 7

Author Response

The manuscript ID: vetsci-2161942, entitled “Iron bioaccessibility and speciation in microalgae used as dog nutrition supplement“ is interesting, but I recommend to improve this rough draft to greater extent that somehow gain better citation game in later time after publication.

  • The Introduction section can be extended by including an adequately argumentation of the research goal.

The introduction has been extended as suggested by the reviewer.

  • Material section, especial Statistic analysis, need to be revised.

Statistical analysis was improved and rewritten in order to be clearer.

The equations from Lines 103 and 128 from Material section need to be numbered and their numbers need to be specified in the manuscript.

The first Equations used in L103 was numbered and cited in the manuscript, while the second one was deleted because it was not relevant.

The Statistical analysis section needs to be reformulated in order to be clearer. See the example from Other remarks.

Statistical analysis was improved and rewritten in order to be clearer from L137 to 147.

  • The results and the discussions need to be more detailed. Please complete the Results and discussions section with more information if the results obtained are in agreement/disagreement with those from other recent studies.

In the submitted manuscript, the authors have already reported if the results were in agreement/disagreement with those from other recent studies in L158-177 for the iron content, L208-224 for iron bioaccessibility, L243-249, L270-274, L294-302, L306-307 for Iron speciation and SEC.

  • Extensive English editing is required.

The manuscript was revised by a certified professor of English specialized in scientific translations

Other remarks: All these remarks have been met

Line 28-30: (P=0.004) need to be inserted immediately after “Significant differences“

Line 110: … for 10 min at 38 kHz … instead of … for 10 minutes at 38 kHertz …

Line 131-132, 135-136: Please reformulate! For example: Kruskal–Wallis one-way ANOVA with Dunn’s multiple comparisons were performed for data with unequal variances. The normally distributed data were compared using a one-way ANOVA with Dunnett’s multiple comparison test.

Line 138-139: … using Rstudio and R 4.2.1 version run in console … Please reformulate!

Line 183, 193: Table 1 instead of table 1

Line 235: Figure 2 instead of figure 2

Line 246: Figures 3 and 4 instead of figures 3 and 4

Line 248: Figures 5, 6, and 7 instead of figures 5, 6, and 7

Line 257: Figure 3 instead of figure 3

Line 269: Figure 4 instead of figure 4

Line 281: Figure 5 instead of figure 5

Line 282: Figures 3 and 4 instead of figures 3 and 4

Line 300: Figure 7 instead of figure 7

Round 2

Reviewer 2 Report

The paper was improved as suggested.

Reviewer 3 Report

Thank you for addressing the given comments.